# “People Will Continue to Suffer If the Virus Is Around”: A Qualitative Analysis of Sub-Saharan African Children’s Experiences during the COVID-19 Pandemic

**DOI:** 10.3390/ijerph18115618

**Published:** 2021-05-25

**Authors:** Samantha Watters Kallander, Rebecca Gordon, Dina L. G. Borzekowski

**Affiliations:** Department of Behavioral and Community Health, School of Public Health, University of Maryland, College Park, MD 20742, USA; samurai7@umd.edu (S.W.K.); rebeccagordon117@gmail.com (R.G.)

**Keywords:** COVID-19, qualitative analysis, child development, health communication, sub-Saharan Africa

## Abstract

Children are particularly impressionable and at risk during a global public health crisis, making it important to examine their unique perspectives. To hear and understand sub-Saharan African children’s experiences with the COVID-19 pandemic, we conducted an exploratory qualitative analysis based on interviews with 51 children, ages 9 to 13, from Nigeria, Tanzania, and Sierra Leone. Applying the organization of Bronfenbrenner’s ecological systems theory, we reveal how COVID-19 affected children’s daily lives and domestic challenges, schooling and neighborhood issues, media use (and its relationship to knowledge and fear of the disease), perceptions of the country and government response, and thoughts of religion and hope. Children’s responses differed greatly, but patterns emerged across sex, age, household size, religion, and country. This study offers guidance and recommendations for meeting the needs of children, especially in times of crisis.

## 1. Introduction

On 11 March 2020, the World Health Organization (WHO) declared the novel coronavirus (COVID-19) outbreak a global pandemic [1]. Given the pathogen’s exponential growth and the absence of a vaccine, there was great concern over the capacity to deal with severe and critical illness. In response, governments enacted proactive containment and measured mitigation strategies. By late March, countries like China had begun taking a hardline elimination approach to the virus, while others started shutting down schools in an effort to “flatten the curve” of COVID-19 cases [2].

Countries around the world differed in their responses to the COVID-19 pandemic, affecting transmission, illness, and death rates. In sub-Saharan Africa, countries faced distinct issues associated with history, leadership, differing political goals, resource allocation, infrastructural challenges, and economic burden, affecting even the youngest of these populations with little control over their own situations [3]. Hearing the voices of Tanzanian, Nigerian, and Sierra Leonean children offers important exploratory insight across three distinct regions of sub-Saharan Africa. Sierra Leone, Nigeria, and Tanzania took very different approaches to the COVID-19 pandemic. It is fair to expect citizens of each country to face this pandemic in particular and dissimilar ways.

### 1.1. Sierra Leone

As of 5 July 2020, Sierra Leone reported 1533 confirmed cases and 62 deaths from COVID-19 [4]. Case numbers continued to grow, and the government of Sierra Leone drew from lessons learned during the Ebola virus outbreak. When Ebola hit West Africa in 2014–2015, the Sierra Leonean government and populace came to understand the benefits of public health control measures [5]. Because of this history, the country clamped down hard and was one of the last nations in the world to confirm a case of COVID-19 [6]. The World Bank provided $7.5 million to Sierra Leone for emergency preparedness relief given the country’s limited resources to handle the current outbreak, including testing kits, medical supplies, and isolation resources for those with confirmed cases [7]. The country closed borders, restricted travel, shut down airports, implemented a curfew, shut down schools, closed gathering places, and announced short-term periods of lockdown where citizens could not leave their homes. Poverty is widespread in Sierra Leone, and the majority of the populace lacked the financial means to handle prolonged stay-at-home orders. Sierra Leonean leaders took an aggressive approach offering resources, communicating health protocols, and employing channels and emergency lines previously used during the Ebola outbreak [8].

### 1.2. Nigeria

In March 2020, Nigeria’s federal government created a task force to fight COVID-19 and help curb the spread of the virus across the country. However, response to the pandemic varied largely across the country’s 37 states. Some Nigerian states, including Kaduna, took directives seriously; others, such as Kano, did not [9]. Nigeria’s historically underfunded healthcare system was slow to implement strong national regulations to manage the COVID-19 outbreak [10]. Political and ideological priorities across the various states of Nigeria revealed additional social and political inequalities and unrest, with the wealthiest communities faring the best during lockdown conditions [10,11]. While Nigeria has the strongest economy in Africa, there is a large economic divide across the country, leaving many poverty-stricken households at great risk during such a crisis [11]. When the five-week lockdown went into effect on 30 March 2020 in the capital city of Abuja and the neighboring states of Lagos and Ogun [9], food scarcity was a great concern with lasting ramifications [12]. On 8 April 2020, the Nigerian government announced that it would distribute 77,000 metric tons of food to vulnerable households and communities struggling with the lockdown [11]; however, it is important to note that these food stores were never distributed and were the partial cause of substantial looting of government storage facilities in central Nigeria for supplies during the pandemic [13]. One of the worst COVID-19 hotspots in Africa may have been Kano, Nigeria’s second-largest city. This area had fewer community restrictions with little social distancing or testing. Officials in this region attributed the increased number of deaths to other causes or labeled them “mysterious deaths” without mentioning coronavirus [14]. Nigeria’s inconsistent actions and economic divides made the toll of the outbreak quite severe, with 28,167 confirmed cases and 634 deaths as of 5 July 2020 [14,15].

### 1.3. Tanzania

The Tanzanian government has not reported new data on COVID-19 cases since the end of April 2020 [16]; at that time, there were 509 confirmed cases and 21 deaths reported [17]. The ruling party in Tanzania has taken counterintuitive steps in handling the pandemic, with its inaction diverging from what is occurring in most of the world’s nations. As of late May 2020, only 652 COVID-19 tests were completed in Tanzania; in contrast, 82,271 and 59,620 tests had been administered in the neighboring countries of Uganda and Kenya, respectively [16]. The then-Tanzanian president, John Magufuli, controlled health information on COVID-19 and is on record encouraging citizens to withhold information, supposedly to protect the country and fellow Tanzanians. Magufuli described COVID-19 as a conspiracy theory and rejected scientific evidence [16]. The WHO criticized President Magufuli for publicly speaking out against the COVID-19 testing protocols, encouraging Tanzanian citizens to continue attending places of worship, and refusing to shut down public transportation [18]. Magufuli had said, “We are not closing places of worship. That’s where there is true healing. Corona is the devil, and it cannot survive in the body of Jesus.” This approach to a public health crisis was not new for Tanzania; reportedly, the government withheld, misrepresented, and underreported cases of Ebola in 2019 [18].

### 1.4. Child Health during COVID-19

Our research team is particularly interested in children’s experiences during a public health crisis like COVID-19, as children have less control over the situations they encounter. Additionally, childhood is a time of great transition with profound cognitive, emotional, and social development. Disasters such as this pandemic can greatly impact children’s current and future lives; [19] unfortunately, COVID-19 may not be the last manmade or natural disaster these children will face. It is helpful for psychologists, educators, communicators, and policy makers to better understand children’s experiences with diseases like this [20,21]. Children under the age of 13 seem to be a significant stressor for households during the pandemic [22], with children in early adolescence (9 to 13 years) being especially aware of external stressors and more at risk of neglect [23]. A qualitative study in this age group can offer guidance for experts who make recommendations affecting population health and well-being, and especially for children who live in challenging conditions.

### 1.5. Ecological Systems Theory

Children’s reactions to difficult events are complex and influenced by multiple factors. Bronfenbrenner’s ecological systems theory (also known as the socio-ecological model) [24] offers an approach to assess the factors and dimensions affecting children within particular circumstances. It has been used in recent literature to categorize multiple levels of childhood interaction with public health issues [25]. The theory describes four levels affecting behaviors: individual characteristics (sex, age), the microsystem (interpersonal interactions such as with family, teachers, and friends), the mesosystem (neighborhood, school, and community groups), and the exosystem (economics, culture, politics, and mass media) [26]. These various levels are fluid and interact with each other; it is important to note that these influences, particularly media, can impact children at different system levels. Both directly and indirectly, system factors influence children’s thinking, perceptions, and behaviors.

In our study, researchers used Bronfenbrenner’s ecological systems theory to organize and give additional context to barriers and facilitators at each level. Using this theory to provide structural insight, lessons learned from this work can inform how to better address the needs of younger cohorts when future manmade and natural disasters occur. The aim of this study was to better understand how sub-Saharan children aged 9 to 13 years experienced the COVID-19 pandemic. Our team posed the following research questions: How did children in regions of three distinct sub-Saharan countries experience the COVID-19 public health crisis? What did children know about the illness and protections against it, and what role has media played in spreading knowledge or fear? How did children’s experiences and concerns vary by gender, age, household size, religion, and country?

## 2. Methods

To address the above research questions of interest, our team conducted an exploratory qualitative analysis with a phenomenological approach, which allows for the analysis of shared experiences around a phenomenon of interest [27]. This approach allowed us to gather themes of the collective experience of the COVID-19 pandemic among sub-Saharan children, while providing a format for comparison of these experiences between sampling regions. By capturing their words, their experiences are given deeper meaning and context through the ecological systems theory.

### 2.1. Ethics

The team submitted protocols and instruments, gaining approval to conduct this research from the University of Maryland Institutional Review Board. Parents and children provided active consent and assent, respectively.

### 2.2. Data Collection

Our team used in-country research teams to collect data. In each country, researchers were in periurban locations outside of the cities of Freetown, Sierra Leone; Kano, Nigeria; and Arusha, Tanzania. For this study, we asked community leaders (e.g., school principals, clergy) for the names of at least 100 children in the targeted age range of 9 to 13 years. These leaders identified boys and girls who would feel comfortable speaking with researchers. Most children came from low-income households, but inclusion criteria required that the participants had access to a mobile phone. No more than one child was sampled from a particular household.

A team of public health and child development experts designed a structured interview guide, which was translated by our in-country research teams. A single in-country interviewer conducted all interviews for a particular region throughout May 2020. Most interviews were done over the telephone to respect social distancing. Consent procedures and interviews were audio recorded. From the community leaders’ list of potential participants, the researchers randomly selected a phone number. He or she called the phone number and, with the child’s parent or guardian, described the protocols, explained the study purpose, and discussed potential risks and benefits. If the parent or guardian gave consent, the researcher then replicated the process with the child to obtain assent.

In Sierra Leone, the researcher contacted 26 parents to find 15 that would provide consent. Those refusing participation expressed concern over their child being audio recorded. Despite assurances to the contrary, refusers worried that obtained information from this study would be turned over to the government. In Nigeria, just one parent refused consent. This father felt the COVID-19 health crisis was not a pandemic, but rather a political conspiracy. No contacted parents in Tanzania refused consent.

Telephone interviews with the participating children lasted, on average, 25 min. The researcher did the interview in the child’s preferred language. Adhering to the structured interview guide, the researcher posed many questions concerning how things have changed during the pandemic, including but not limited to the following: What was life like (and what had changed) during the COVID-19 public health crisis? Have things been easier or more difficult, and why? Do boys and girls have the same or different experiences during the pandemic? What was the child’s media use during the crisis? What knowledge did they have about the virus and protections against it? What were the child’s concerns related to COVID-19? What has their government done or told them about the crisis? What lessons can we learn for the future, and how will things be different? Audio-recorded interviews were transcribed and translated into English by the local research teams. Electronic audio and translated documents were de-identified and securely delivered to our team for coding and analysis.

### 2.3. Participants

The final study sample (Table 1) consisted of 20 children from Nigeria, 15 from Sierra Leone, and 16 from Tanzania, exceeding the sampling goals of 15 per region. There were similar numbers of boys and girls, and roughly 20% of the sample was in each of the five age categories. Around half of the interviewed children came from small households, defined as six or fewer people living in the home; the remaining half lived in large households, with seven or more people in the home. About half of the children came from Muslim households, while the other half were Christian.

### 2.4. Data Analysis

To stay true to the participants’ words and experiences, two independent coders read each transcript and used grounded theory coding techniques to gather all emerging categories from the text [27]. The grounded theory coding started with a line-by-line review of the text, creating open codes that captured pieces of text relevant to the research questions. The open codes were then gathered into focused codes to capture major coding themes emerging from the interviews. Finally, these categories were reorganized and revised to reach a broader understanding of the patterns across all three regions. Coders used NVivo 12™ software to track and categorize codes. The researchers translated the focused codes into broader themes and subthemes to best represent study participants’ perceptions of life during COVID-19 and capture any phenomenon. Additional analysis was conducted to see how children’s experiences varied by gender, age, household size, religion, and country.

### 2.5. Data Quality

Consistent sampling was achieved across all five ages, both genders, and religious affiliations. An international team of subject matter experts developed the interview guide, and it was tested with child informants to make sure children could understand and respond to the questions. In-country teams reviewed, translated, and reverse translated the guide for accuracy and clarity, while also ensuring cultural sensitivity.

Researchers audio recorded the interviews. During the data collection process, the design involved using a single researcher conducting all the interviews, which allowed for quality control and consistency. The principal investigator trained the researchers to follow the interview guide, adhering closely to directions and exact questions, and discussing each question with interviewers in detail to ensure understanding of the process. The guide was pretested with interviewers, and all interviews were audio recorded and reviewed to further ensure fidelity. The researchers conducted the interviews in the child’s preferred language, transcribed the responses in direct context of the interview guide, and then translated the transcripts into English. The researcher who conducted the interviews also did the transcriptions and translation to best capture the true sentiment of the child’s words and experiences.

During data analysis, coders consulted with the in-country interviewers as needed to ensure full comprehension of the data. Each coder completed the process separately, reflecting on personal judgments and preconceptions throughout the coding and thematic synthesis process. Peer debriefing occurred in order to share findings between the coders and control bias in the coding process.

## 3. Results

Thematic analysis revealed five unique coding themes to capture changes during the pandemic, each with several subthemes (Figure 1). The ecological systems theory also offers a structure to the results, highlighting how different spheres of influence affect children’s perceptions of COVID-19 while identifying key barriers and facilitators to their lives and health identified by children in relation to public health crises like this one (Table 2). Within each level of the theory, we note similarities and differences, especially by participants’ demographic variables and country. Religion emerged as both a demographic variable and a prominent coding category, suggesting the important role it plays in both personal and community life. To exemplify themes, we provide italicized key quotes from the interviewed children, as well as tabulations of comments to give context to the results and further inform comparisons.

### 3.1. Individual Characteristics

Younger children in the study, particularly the 9-year-olds, were the least talkative age group, providing an average of 72 unique open codes per transcript. This was less than all other age categories, where children provided on average between 85 and 91 open codes per transcript. The 11-year-olds were the most talkative and eager to share. Boys tended to be more talkative than girls; boys produced an average of 89 open codes per transcript, while girls produced an average of 81 open codes per transcript.

Based on their comments, we believe that girls and boys had similar experiences with the COVID-19 pandemic, with only a few variations of note. In fact, when the interviewer directly asked participants if they felt boys and girls had different experiences, 40 of the 51 children stated that boys’ and girls’ lives were similar during COVID-19.

Children of different ages reported similar experiences during the COVID-19 pandemic with the youngest and oldest children differing the most. In contrast, household size affected the COVID-19 experience. Children from larger households (18 of 23, 42 mentions) more frequently described food insecurity than those from smaller households (8 of 24, 23 mentions). Chores were also mentioned less frequently by those from smaller compared to larger households.

Differences at the individual level were also addressed throughout the five themes (Figure 1) in the microsystem, mesosystem, and exosystem where noteworthy. In this study, the most prominent themes in our interviews emerged at higher levels of the ecological systems theory, especially at the exosystem level. To organize the rest of the findings, we started with the microsystem level.

### 3.2. Microsystem

Within the ecological systems theory’s second level, the microsystem, one considers interpersonal relationships and connections among family and friends and in the home settings [26]. Participating children discussed interpersonal relationships and connections among family and friends in the home under “Challenges in Domestic Life” as depicted in Figure 1.


*“It has become difficult because everything has changed for me.”*
(Boy, Age 9, Nigeria)

When describing daily life, 21 of 51 participants expressed concerns about being bored at home during the pandemic. This was more commonly discussed among children from Nigeria and Sierra Leone than children from Tanzania. The daily activity mentioned most often by children was doing chores, followed by studying, interacting with family, using media all day, bathing, praying, and playing. These activities were fairly equally distributed among regions, with the exception of daily prayer, which was reported more often by Sierra Leonean (8 of 15) and Nigerian (6 of 20) than Tanzanian children (3 of 16). Children from Nigeria discussed studying less than those from other regions. Nigerian children also talked more frequently about family time. Younger children (9-year-olds) did the least praying and fewer chores. Older children (13-year-olds) had the greatest daily media use. Additionally, those from small households spoke more about chores, while those from larger households talked more about engaging in daily prayer and religious connections. Muslim children mentioned spending more time with family on a daily basis and more time seeing friends, while Christian children talked more frequently about studying. Children across all age groups expressed that their days all looked the same during the COVID-19 pandemic.


*“The same thing every day.”*
(Boy, Age 10, Tanzania)

Among the 51 participants, 14 expressed that they could not see their friends, with this concern being voiced most often among Sierra Leonean children. A different set of six children mentioned seeing their friends anyway, despite restrictions. Seeing or not seeing friends seemed to be a significant stressor, with more boys (12 of 26, 16 mentions) expressing this than girls (8 of 25, 9 mentions).


*“I don’t have any child my age to talk to or play with.”*
(Boy, Age 11, Sierra Leone)

Children described family dynamics as both a barrier and a facilitator. Out of the 51 participants, 8 described their parents’ level of worry. Girls (7 of 25, 10 mentions) talked about this far more often than boys (1 of 26, 1 mention).


*“My dad will go to work and most times he will grumble that things are now harder than before.”*
(Girl, Age 12, Sierra Leone)


*“Before, my family was very happy family. Now that things are difficult, my family is not happy.”*
(Girl, Age 9, Sierra Leone)


*“My parents…I make them happy so they don’t get angry, and I work calmly.”*
(Girl, Age 11, Tanzania)

In Sierra Leone, children frequently mentioned “hiding” from their family and parents (8 of 15). Most of these comments came from children of larger households (6 of 10) compared to those from smaller households (2 of 5). “Hiding” was not mentioned by children from other regions. 


*“I will hide from everyone in the house to study my school notes…I will always hide from my parents.”*
(Girl, Age 10, Sierra Leone)

Nigerian children more commonly expressed helping their parents at home (10 of 20). Several children described that engaging in such activities with family members was a coping mechanism; a few even remarked that this was a positive consequence of being at home.


*“I manage by sitting down with parents and discuss.”*
(Girl, Age 13, Tanzania)


*“Before, my mom don’t have time for me. But now, I am surprised she has lots of concerns for me since this pandemic. And most importantly, she is giving the mother’s love I was yearning for a long time.”*
(Girl, Age 12, Sierra Leone)

Many interviewed children talked about the comfort of their household, with 14 of 51 participants talking overtly about crowding. Most Sierra Leonean children raised this issue (11 of 15, 30 mentions) and just a few Nigerian children (3 of 20, 5 mentions). No Tanzanian children raised this issue. Across the regions, those from larger households (10 of 23, 27 mentions) talked more about comfort than those from smaller households (4 of 24, 8 mentions), discussing noise, size of their homes, and crowding.


*“When we have all the elderly people around the house lots of argument and noise so I have to find a corner in the house and hide with my books. And study quietly.”*
(Boy, Age 13, Sierra Leone)

Stress was a common household problem, with 34 of 51 participants expressing explicit concern about how their family was coping during the pandemic. Again, this was mentioned most frequently among those from Sierra Leone, with all 15 children describing feelings of stress. Only half of the children from the other two regions talked about this. The most overt mentions of mental health concerns came from Tanzanian participants, with one 9-year-old girl saying, “I manage badly.” The oldest children most frequently commented about stress, as did those from the larger households.


*“I lose the interest on studying, I find myself lonely, unlike I used to be.”*
(Boy, Age 11, Tanzania)


*“I am unhappy.”*
(Girl, Age 9, Nigeria)

### 3.3. Mesosystem

We examined children’s relationships with their community and organizations within the community, with a focus on school, neighborhood, and church in “School and Neighborhood Changes” as depicted in Figure 1.


*“I miss school classes, to be educated.”*
(Boy, Age 11, Tanzania)

Most children in this study worried about missing school, with 42 of 51 children overtly mentioning problems related to being out of school. This issue was repeatedly mentioned within and across borders; among these 42 participants, there were 158 separate mentions. All Sierra Leonean participants talked about missing school (15 of 15, 55 mentions), while three-quarters of the Nigerian (15 of 20, 43 mentions) and Tanzanian (12 of 16, 60 mentions) children expressed this worry. In Tanzania, children did not talk about religious schools, while 7 of the 20 Nigerian children talked about missing their Islamic school and 3 of the 15 Sierra Leonean children mentioned being away from their Christian-affiliated school.


*“I’m not happy when schools are closed.”*
(Girl, Age 10, Sierra Leone)


*“What has changed is that, learning as we used to does not exist, you are learning on your own that create stress, it affects the brain.”*
(Girl, Age 13, Tanzania)


*“Yes, I’m concerned about my inability to go to school and also Islamic school.”*
(Boy, Age 13, Nigeria)

The majority of the children (28 of 51) talked about studying at home as a common activity and part of their daily routine. Around 10 children discussed how media use was part of their education. By region, children most frequently discussed this topic in Tanzania (13 of 16) and Sierra Leone (11 of 15). Fewer Nigerian children talked about daily studies (4 of 20). Students from Sierra Leone discussed studying online and how their government facilitated this process, while those from Tanzania mentioned how they were watching public access educational media.


*“Government has supplied books to schools so that kid will have these books to study at home. And online classes for kids.”*
(Boy, Age 10, Sierra Leone)


*“I watch Ubongo Kids.”*
(Boy, Age 10, Tanzania)

When discussing neighborhood engagement, 16 of 51 participants expressed frustration that citizens could no longer work or visit local markets, with the majority of these comments coming from Nigerian children (12 of 20). Boys (10 of 26, 14 mentions) seemed more frustrated than girls (6 of 25, 8 mentions) did about this issue, more frequently mentioning it as a concern. Not only did Nigerian children talk about neighborhood constraints, but also they seemed to be the least worried about interacting with community members. Considering the comments of Tanzanian children on this issue, these children seemed to be more concerned about the problems associated with interpersonal interactions.


*“You might go out, find a group of people sitting, you don’t know maybe one of them has corona, and those who sit with him they don’t know, and you are passing by, maybe among them one of the person has corona, and you get a problem.”*
(Girl, Age 10, Tanzania)

We found that 12 of the 51 interviewed children expressed positive lessons and changes for the future. Children were mostly optimistic across all regions, but Tanzanian children voiced skepticism related to their future educational paths. 


*“I was thinking of being a lawyer in the future, but COVID-19 has inspired me to become a medical doctor in my future career to save lives.”*
(Girl, Age 12, Sierra Leone)


*“I don’t know how about schools, when we get back to them, how they will look like.”*
(Boy, Age 12, Tanzania)

### 3.4. Exosystem

Many of the children focused on issues related to the exosystem level, the highest level of the ecological systems theory. We examine issues of social context, culture, economics, and politics in “Media Impacts and Changes; Country and Government Actions, Health Protocols, and Neglect; and Religion and Hope” as depicted in Figure 1.

#### 3.4.1. Media Impacts and Changes


*“I saw a film where people were running away from the person that has coronavirus, that they should give 2 m distance, and people were locking their houses when they see him.”*
(Girl, Age 13, Nigeria)

Most participants reported home access to a television (41 of 51) and a radio (40 of 51). Nigerian children had the least access to TV (14 of 20), and Tanzanian children had the least access to radio (9 of 16). Children described mobile phones as the next most common additional media item (22 of 51). Sierra Leonean children reported the most access to mobile phones (10 of 15), while Nigerian children had the least (3 of 20). 

Regarding media use, 48 of the 51 participants said they had moderate (near daily) to high (daily) media use, with about half using media daily. While moderate to high media use was consistent across regions, the Sierra Leonean children reported the greatest use rates, followed by the Tanzanian and then Nigerian children. For example, in Sierra Leone, just 1 of the 15 children offered that his daily use was moderate while the other 14 indicated high daily media use. In contrast, among the 20 Nigerian children, 11 described moderate media use (using media almost daily) and 7 had high media use. 

While many children expressed that media use was a regular part of their lives, several explained that they were spending more time that usual using media. The type of use had changed for some; during the COVID-19 pandemic, they were more frequently using educational media, as well as watching or listening to news programs. The high and increased media usage was consistent across participants from the three regions. 


*“Before coronavirus, I was not frequently using media like I used to now. I use media now more than before to get updates on this deadly COVID-19.”*
(Boy, Age 11, Sierra Leone)

The children’s comments revealed a complex relationship between media use, fear, and knowledge. Regarding fear, slightly more than half of the children (28 of 51) reported being very or highly worried about COVID-19 in general, and 31 of 51 children were specifically fearful that someone they know might die of the virus. Sierra Leonean children seemed most concerned, with all 15 children saying they were very worried, 14 saying that a family member might get sick, and 12 thinking that someone they know could die. Tanzanian children were less fearful, with 9 of 16 children expressing high levels of worry. Nigerian children were the least fearful; just 6 of 20 children said they were highly worried. Boys and girls were equally fearful, but 9-year-olds reported the least fear of any age category. 

Children reported what frightened them most about COVID-19. An 11-year-old Sierra Leonean boy said, “They cannot tell who has coronavirus or not.” A 9-year-old Tanzanian girl commented, “I watch news every day, I always see how people died in corona epidemic.” A 10-year-old Nigerian boy talked of how scary the virus was, attributing his fear to “the way the virus looks. It looks like round with pointed things around it.”

Media seemed to be the source of fearful messages in this study. From the interviews, we observed that children with the highest fear also had the greatest media use, regardless of region; 29 of 30 with high fear also had high media use. The reverse was not true, however. Not every child with high media use also had high fear; we found that 29 of 48 children with high media use expressed high levels of fear.

Around half (27 of 51) of the interviewed children had a high level of knowledge about COVID-19. Boys more so than girls seemed to have greater knowledge of symptoms. Almost all children knew that the first reported case occurred in China; just 4 children (all 9-year-olds) lacked this information.

Most children (38 of 51) reported that they had acquired their knowledge about COVID-19 from media sources as opposed to getting this information from someone in their lives (8 of 51). However, despite no great differences in children’s overall media usage across the three regions, knowledge levels differed. All 15 participants from Sierra Leone had high knowledge of COVID-19, answering most questions about the virus correctly. Comparatively, smaller numbers of Nigerian (8 of 20) and Tanzanian (4 of 16) children had high knowledge. 


*“Things I remembered are when they talk on SLBC TV that the President of Sierra Leone has closed all border crossing to the country, people should stay at home and then at 21:00, everybody should stay indoors curfew.”*
(Boy, Age 11, Sierra Leone)


*“On TV, they show you can use lemon juice to boil and drink, or you take lemon leaves, you take the flu medicine you put in the water and the guava leaves, then you put on the stove boiling, and then you put the water in the basin after getting really hot, you take a blanket, you cover your body until the sweat comes out, if you get flu or cough, you do it every morning.”*
(Boy, Age 12, Tanzania)

This study considered the relationship between knowledge and fear. In Nigeria, for example, half the children with high knowledge also had high levels of fear (4 of 8); however, of those with high fear (6), 4 had high knowledge. Additionally, almost all of those from Nigeria with high knowledge also had high media use (7 of 8). Alternatively, all Sierra Leone participants had high fear, high knowledge, and high media use. In Tanzania, just 4 of 16 participants had high knowledge, but 3 of these 4 also had high fear, and all 4 had high media use.

Children discussed lessons learned from the media. Most children (45 of 51) described media’s beneficial effects, expressing that they were learning ways to protect their health and methods for handling difficult situations. This was consistent across regions.


*“Wash hands regularly will reduce the risk of getting other virus in the future.”*
(Boy, Age 12, Sierra Leone)


*“I have learned to watch the media.”*
(Girl, Age 13, Tanzania)


*“The message I have is to tell people about this corona pandemic and how it has destroyed the world. To observe all health protocols like wash your hands, social distancing, avoid crowded areas and the use of facemask.”*
(Boy, Age 11, Sierra Leone)

#### 3.4.2. Country and Government Actions, Health Protocols, and Neglect


*“Follow the rules and regulations of the government on the health protocols for COVID-19. Even though the health protocols may worry them, but they should abide to it. Nobody is above the law.”*
(Boy, Age 11, Sierra Leone)

Food insecurity has to do with children’s perceptions of food access and affordability during the pandemic. Just over half of the interviewed children (27 of 51, 66 mentions) discussed general difficulties of getting or buying food. All 15 Sierra Leonean children mentioned this issue (38 mentions), compared to 11 of the 20 Nigerian children (27 mentions). Just one of the 16 Tanzanian children commented about this topic. Among those interviewed, those from larger households (with presumably more mouths to feed) raised the issue of food insecurity (18 of 23, 43 mentions), more so than those from smaller households (8 of 24, 23 mentions). Muslim children mentioned food insecurity more frequently (16 of 25, 38 mentions) than Christian children (10 of 26, 27 mentions). 


*“In Sierra Leone, not everybody can afford to eat. If you have money, you are not worried at all. But we, my family, get worried when they announced lockdown. It is difficult for my family.”*
(Girl, Age 13, Sierra Leone)

Children from the three regions described issues related to struggling economies (12 of 20 Nigerian children, 16 mentions; 7 of 15 Sierra Leonean children, 21 mentions; 8 of 16 Tanzanian children, 12 mentions). Just over half of the children (27 of 51, 49 mentions) discussed how their parents lacked work, but often framed this issue as a community problem. The Tanzanian children made the most explicit mentions about the national economy. 


*“The economy is not stable because of corona, and Tanzania has no direction.”*
(Boy, Age 11, Tanzania)

In our interviews, 9-year-olds talked the least about the struggling economy (3 of 11, 7 mentions). Boys (17 of 26, 26 mentions) talked about economic issues more frequently than girls (10 of 25, 12 mentions). Muslim children mentioned economic issues more frequently (15 of 25, 27 mentions) than Christian children (12 of 26, 22 mentions). This issue was raised with similar rates among those from different-sized households (larger—13 of 23, 22 mentions; smaller—14 of 24, 27 mentions). 


*“Things are difficult because prices of commodities have increased and there isn’t enough money to buy things…It’s easier for kids from wealthy families and more difficult for us from poor families.”*
(Boy, Age 12, Nigeria)

Boys (19 of 26) talked more overtly about their knowledge of government actions to combat COVID-19 than girls did (15 of 25). Children from larger households (19 of 23) also seemed to have more knowledge of government actions compared to those from smaller households (15 of 24).

In these interviews, 34 children made comments about what their governments were doing, mostly mentioning stay-at-home and lockdown orders, health protocols, closing schools and churches, restricted travel, and curfew orders. Occasionally, children talked about government provisions or sanctions for not obeying regulations. All 15 Sierra Leonean children described government actions, compare to just half of the Nigerian and Tanzanian children. Several of the Sierra Leonean children discussed distribution of provisions, describing how the government was offering Veronica Buckets for handwashing in the home, as well as school and study materials. Some (3) mentioned the government’s emergency phone line for reporting symptoms. Among Nigerian participants, children mostly described government restrictions, such as stay at home orders and school and church closures. A 13-year-old Nigerian girl talked about the government providing personal protective equipment (PPE) and rice; however, she commented that “The government said they were going to distribute face masks and rice, but when we went out and saw where they kept the rice, the rain had damaged the rice.” The Tanzania children made fewer mentions of specific rules and restrictions, with most comments focusing on health protocols and staying at home. 


*“President allows things…sports to proceed, schools will soon be opened...on 1st June university students will go back to school, together with people not locking themselves inside, they should keep working, and not be afraid of corona.”*
(Girl, Age 13, Tanzania)

Country differences emerged when participants discussed compliance with health versus government officials. Such comments often focused on government trust and criticism. Despite some complaints, 34 of the 51 participants overtly promoted listening to the government, while a quarter of the sample explicitly mentioned obeying government rules and regulations. Sierra Leonean children talked about obeying the government instructions (6 of 15), whereas Nigerian children mentioned following health protocols described by doctors (17 of 20). Tanzanian participants were most likely to stress personal protection measures from the COVID-19 virus (11 of 16).


*“Follow the rules and regulations of the government.”*
(Boy, Age 11, Sierra Leone)


*“People should obey the instructions of doctors.”*
(Girl, Age 9, Nigeria)


*“Protect themself from corona, because if you do not protect yourself you will die.”*
(Boy, Age 10, Tanzania)

Nigerian children (7 of 20) criticized their government with the highest rates. In contrast, Tanzanian children most often provided comments suggesting trust in their government (8 of 16) as well as national collectivism (4 of 16).


*“People have…people to be motivated…to give priority to Tanzania, so that people continue to have...people hoping that they will be a day when people will recover.”*
(Boy, Age 11, Tanzania)


*“I have heard that the government has said that schools will be opened and children will continue to study and that corona will decline, when the corona increase you are not allowed to say how many have recovered, how many have died and how many have infected. I’m thinking about that is true, because he (the President) is saying that to us as his fellow Tanzanians brothers and sisters, his colleagues or relatives and friends.”*
(Girl, Age 11, Tanzania)

Interviewed boys expressed greater trust in their governments (12 of 26), while girls were more likely to offer criticism (7 of 25). However, more girls (6 of 25, 8 mentions) expressed clear messages of national collectivism than boys (3 of 26, 5 mentions). Children from smaller households were more likely to overtly mention obeying the government (9 of 24, 11 mentions) or protecting oneself (9 of 24, 13 mentions), whereas those from large households were more likely to cite adherence to health protocols (20 of 23, 22 mentions). Additionally, Muslim children (22 of 25, 23 mentions) were more likely to mention listening to health protocols than Christian children (12 of 26, 13 mentions), while Christian children (10 of 26, 12 mentions) were more likely to mention obeying the government than Muslim children (3 of 25, 4 mentions). Muslim children were also more apt to criticize their country officials (8 of 25), while Christian children expressed trust in their government (13 of 26).

On the topic of countrywide susceptibility and severity, 19 of 51 children suggested that there was less disease in their home country than other countries. Around half of the Sierra Leonean and Tanzanian children specifically said this. Most children (36 of 51) perceived COVID-19 to be serious and severe, with all of the Sierra Leonean children mentioning this.


*“Things are easier here…the number cases are fewer compared to other countries and also the number of deaths are lower.”*
(Boy, Age 13, Nigeria)

#### 3.4.3. Religion and Hope


*“Protect ourselves from this virus and we believe that God will remove this…work will continue as before corona arrives.”*
(Girl, Age 11, Tanzania)

Children expressed religious connections in different ways. Fifteen of 51 participants explicitly mentioned prayer as a valuable tool to better manage and improve things during this time. Boys (10 of 26, 15 mentions) made such comments more frequently than girls did (6 of 25, 7 mentions). Participants from larger households (9 of 23, 12 mentions) seemed to more frequently talk about religion than those from smaller households (7 of 24, 10 mentions). Similar numbers of children across the three regions discussed religion, but content varied. Nigerian children (16 of 20 identifying as Muslim) mentioned Islamic schools, while Sierra Leonean children (8 of 15 identifying as Christian) mentioned Christian schools. Sierra Leoneans most often discussed that physical churches had closed; Nigerian and Tanzanian children were more likely to talk about how they used technology to continue religious practice at home.


*“When I wake up from bed, I kneel down and pray to God almighty.”*
(Boy, Age 10, Sierra Leone)


*“I thank God for my life.”*
(Girl, Age 13, Sierra Leone)


*“People should pray to God for the remedy, and they should also obey all the rules and regulations that has been set.”*
(Boy, Age 12, Nigeria)

Only one participant in the entire study, a 13-year-old boy from Nigeria, described the pandemic as a time of religious judgment. He said, “People should pray and repent.”

Several children (15 of 51) offered hopeful messages for the future. This was particularly common among Tanzanian children (8 of 16), and fairly common among Nigerian children (6 of 20); it was rare among Sierra Leonean children (1 of 15). Boys (10 of 26) made such comments more so than girls (5 of 25). Muslim and Christian children made hopeful comments at similar rates.


*“I think that, as far as Tanzania is struggling, there is a day we will find a cure and then they will recover and we will continue with the studies and people will continue to work and all schools will be opened.”*
(Boy, Age 11, Tanzania)


*“Protecting themselves from corona, and not be afraid, because we went through many disease, especially young people...like HIV.”*
(Girl, Age 13, Tanzania)


*“Things will change, I just believe so.”*
(Boy, Age 10, Nigeria)

## 4. Discussion

This study offers fascinating information into children’s experiences with COVID-19. The ecological systems theory organizes how critical factors and elements emerged and affected children during this pandemic. While a child’s sex and age at the individual level, household structure at the microsystem level, and community at the mesosystem level all played roles, the exosystem level, including the child’s religion, home country, and his or her exposure to media seemed to have the most striking impact on children’s perceptions of the nature and severity of COVID-19. This level therefore requires more contextual exploration. We follow this with relevant and result-based recommendations at each level of the ecological systems theory.

### 4.1. The Exosystem: Historical and Political Context

The Sierra Leonean children of this study had the highest levels of knowledge, the most understanding of what their government was doing, and the greatest overall adherence to health protocols. These children wholeheartedly believed that “coronavirus is real” and that the country was doing what it needed to control the virus. These children also reported the most media use to learn about COVID-19. We suggest that Sierra Leone’s coordinated control measures and communication efforts [8,28] were creating a successful response, at least in the children in our study. That these children were knowledgeable about public health efforts is likely related to the country’s proactive approach as well as its history with the 2014 Ebola outbreak. During the Ebola pandemic, Sierra Leone had 14,124 cases and 3956 fatalities [29]. To avoid a similar tragedy, the Sierra Leonean government responded quickly and comprehensively to COVID-19. The 9- to 13-year-olds in our study seemed to be benefiting from these priorities and learned lessons.

The Tanzanian government, in contrast, has offered a haphazard and non-serious response to the COVID-19 pandemic [16]. It is not surprising that Tanzanian children in this study were more ignorant and less concerned about this public health crisis. To date, the Tanzanian government has withheld credible information on COVID-19 cases or fatalities; leaders have urged its citizens to continue usual behaviors and keep the country’s economy going [16,18,30,31]. In this study, researchers heard Tanzanian children describe how their leader, President Magufuli, stressed national collectivism. Several children explicitly mentioned how they felt it was inappropriate to discuss countrywide cases. Despite high rates of media use, Tanzanian children had the lowest reported levels of knowledge about COVID-19. This suggests that either children are watching other programming, or the Tanzanian media is not presenting information on the virus in this particular region. The government response and resulting children’s perceptions are extremely problematic during this global public health crisis [32,33,34].

In Nigeria, the inconsistent messaging and dissimilar methods of control across the country manifested in the sampled children showing differential levels of fear and understanding of COVID-19. For the region of Kano sampled, Nigerian children reported less media use than children in the other two countries. Not surprisingly, these same children seemed to have less fear, lower overall knowledge, and little awareness of how their government is responding to the pandemic. The economic and political divides in Nigeria [11]. as well as the mixed approach to controlling the pandemic [9] and lack of lessons learned to bolster their healthcare system from Ebola [10,35], help explain children’s reflections on COVID-19. Without an integrated federal response, the Nigerian states have taken matters into their own hands, resulting in varied responses across the 36 states [9].

Across the three regions, the Nigerian children in our study described the least government trust and were most likely to overtly criticize their officials. In contrast, Tanzanian children expressed the highest levels of trust and least criticism [36]. Given the above discussion of country responses, this is notable. Interestingly, the Nigerian children reported less national collectivism with more trust in health practitioners over government officials. In contrast, Tanzanian children, who had little knowledge and understanding of government, adhered to the President and mass media’s messaging, expressing strong national collectivism and even pride in their country’s isolationism.

In discussing government trust, historical perceptions of political corruption may play a role. Perceptions of government healthcare systems and responses to health crises are associated with government trust [37]. This offers some explanation for this study’s findings. Following its devastating 11-year civil war, the Sierra Leonean government reconfigured and tried to effectively respond to the Ebola crisis [38,39]. Processes in this exosystem improved, making the populace and young participants of this study better equipped to handle this current health crisis. Nearly half of these Sierra Leonean children talked about national collectivism, showing support around the government’s quick actions and concerted educational COVID-19 efforts. Among the Tanzanian sample, half reported government trust, with very little overt government criticism and greater national collectivism. Possibly, these Tanzanian children were accepting President Magufuli’s “man of the people” message and believed not much needed to be done about this virus. Another explanation is that children in this Tanzanian sample might have felt fearful expressing criticism of their government, worrying about President Magufuli’s strong messages to “stamp out” dissention among the population [40]. Finally, Nigeria’s lack of consistent health communication and healthcare efforts may have contributed to the sample’s overall low government trust. The Nigerian children in our study more frequently mentioned a struggling economy. Often, this talk of poverty corresponded with discussing low trust in the government, supporting previous associations in the literature [41]. In our study, the most religious messaging from study participants came from Nigerian children, as did the least government trust, which is also a trend supported in the literature [42].

Another exosystem factor, religion, played a significant role in how children in this study dealt with COVID-19. Many participants explicitly offered religious messages of hope. Children discussed how they missed their religious schooling and services. Muslim children in our study were more likely to report adherence to health officials over obeying the government while also being more likely to criticize the government openly. This aligns with the larger Muslim population in northern Nigeria displaying a lack of trust in government, compared with the larger Christian population in Tanzania displaying loyalty to and trust in their government [43]. In Tanzania, Christian children may be at greater risk with President Magufuli’s advice to attend church services, saying Jesus would protect them from the spread of COVID-19 [18].

Children, especially those from the sampled regions of Sierra Leone and Nigeria, frequently raised the issue of food insecurity. In Sierra Leone, 53% of the nation lives below the poverty line, with a chronic malnutrition and stunting rate of 31.3% [44]. Over 3 million of the country’s almost 8 million citizens lack adequate nutrition [44]. All the Sierra Leonean children in this study talked about food insecurity as a concerning consequence of the COVID-19 lockdown. Nigeria, which is a much more populous country with almost 200 million people, has more economic diversity. Persistent poverty is a problem, with around 110 million or 60% of the population living below the poverty line [45]. Nigeria’s large economic divide means that some will desperately need food assistance, while those who are wealthier will fare better during a lockdown [10]. Over half of the Nigerian children in our study sample mentioned food insecurity, mirroring the country’s poverty rate. Additionally, Nigeria was the only country where several children explicitly mentioned the government’s need to deliver rice to households during the lockdown [11]. Larger households were more likely to be religious and tended to be Muslim. Consequently, it was children from these Muslim families reporting higher rates of food insecurity and economic hardship. In Tanzania, a much smaller fraction (36%) of the populace lives in poverty and a third (32%) of children under five years old are reportedly malnourished [46]. Only one Tanzanian child from our study raised the issue of food insecurity during the COVID-19 pandemic, suggesting that this is either not a significant problem or that Tanzanian children hesitated from mentioning food insecurity as it might be construed as criticizing the government.

### 4.2. Recommendations for Future National Health Issues

The children identified many barriers and facilitators during their experiences with the COVID-19 pandemic. While we recognize that some of these issues deal with structural and cultural challenges that are difficult to address, paying attention to the challenges raised and noting identified support encountered by these children may help better address future health crises. Researchers and in-country teams agreed that those in power should closely consider comments from study participants discussing those in need, prioritizing food for hungry children, and overall high levels of struggle and stress. From this exploratory study, we offer recommendations for handling future manmade and natural disasters, particularly in sub-Saharan Africa.

At the individual level, it is important to address and prioritize knowledge and empowerment among girls. We found that girls (who talked less than boys) worried more about their families and homes, voiced less concern about general economic issues, and seemed less hopeful about the future. This makes sense as, in sub-Saharan Africa, girls more so than boys are expected to do household chores and are at greater risk of not finishing school [47]. Especially during a disaster, efforts should be made to alleviate burden and stress. Girls need to feel empowered to use their voices and be heard so that officials can better protect them and their futures [48,49].

The study’s older children expressed more stress. These children also had more responsibilities and were more knowledgeable about the community’s circumstances than younger children. Government communication should address children’s stress, perhaps adding information and infrastructure to managing stay-at-home directives. We found that stress levels were actually highest in Sierra Leone where the most information was available. This suggests that the nature of communication must change. Media messages should not only focus on sound science-based information that appeals to the audience, but also incorporate hopeful messages to help children cope with the information they are receiving. Lastly, age-appropriate information about personal coping skills can be communicated during stressful times.

At the microsystem level, positive family interactions and dynamics appear protective. The reverse is also true. Sierra Leonean children who lived in crowded households noted more stress and reported regularly hiding from their family members. Community efforts should focus on household and family dynamics, encouraging parents to give their children time to study and cope with various circumstances. Household size was a key indicator of difficulties, with more discomfort reported by those from larger households. While being asked to help out during a crisis can be positive [50], one does not want to overburden those who are not ready to take on such responsibilities.

At the community or mesosystem level, the least studying occurred at home in Nigeria where children rarely mentioned an infrastructure for stay-at-home education. It is essential to plan ways for students to continue their studies during a lockdown, even during a crisis. Governments must strive to lessen children’s worry over missed content. The study’s sample felt high levels of distress over if and how they would continue their education and return to the classroom. Messages to the public should be clear about school plans, supporting practices, and securing the future of children’s educational paths. Additionally, media should take advantage of the opportunity to share more educational programming with children. Effective educational media is available in sub-Saharan Africa; millions of children throughout the continent are currently enjoying and learning from programming such as Akili and Me and Ubongo Kids [51,52].

Finally, on the exosystem level, media must carefully consider how they share risk communication. Media conveys cultural values and perspective, and this study offers clear evidence that impressionable children are part of the general audience. Inconsistent messaging creates knowledge gaps and misconceptions. Media can easily spread fear and stress without providing adequate coping mechanisms. We see that even high-quality information is associated with increased fear and worry. As part of the provided information, messages should assure and offer hope. Among audiences with high media access and use, there is ample opportunity to teach positive lessons for health and increase knowledge. From our data, we saw this was done effectively in Sierra Leone and less so in Nigeria and Tanzania.

The government should work closely with the media to share their actions and encourage openness during crises. There is a relationship between a lack of knowledge and communication on government actions and a lack of trust in or willingness to obey the government, which was particularly prominent among the Nigerian children in this study. While national collectivism is high in Tanzania, political control and government secrecy remain problematic and should be avoided during a global crisis [16,18]. Additionally, only a handful of sampled children talked about government provisions and support. Especially where this has been done [7,11], it may comfort children to know that community actions are occurring. Consistent messaging and transparency throughout the crisis seems to be a key finding that is paramount to children’s experiences during a crisis like this [53].

Finally, prayer seems to be a strong coping strategy during times of crisis, and an effort should be made to allow for safe religious worship regardless of lockdown orders. Evidence shows that faith can greatly improve mental health outcomes and coping for those in situations like the COVID-19 pandemic [54,55,56]. Given the highly religious lives these children lead, it is important to convey religious messages of hope during manmade and natural disasters. In this study, skepticism and confusion were barriers and surveyed children were looking for positive messages, especially through their mosques and churches. However, leaders should make careful efforts to ensure that religious connections and prayer do not contradict information from scientific leaders, as was the case in Tanzania when President Magufuli insisted that prayer would keep COVID-19 away [18]. Messages from religious leaders as opposed to political figures should be prioritized at the community level.

## 5. Strengths and Limitations

This study had drawn on data provided by an exploratory sample from three distinct sub-Saharan regions to make sense of children’s experiences during the COVID-19 pandemic. By sampling across three distinct regions of sub-Saharan Africa, this work allowed for intriguing comparisons and recommendations to be investigated by future research. The methodology and data analysis offered understanding of children’s experiences, while the theoretical framework provided a larger and fascinating contextual lens for the work. Additionally, our in-country teams provided further cultural context underlying the children’s comments.

Study limitations included data collection variations across regions and children. In some cases, interviews occurred by telephone, in others the interviews happened face-to-face with both researcher and child wearing masks. In both situations, interviewers could not see children’s facial expressions. Challenges in collecting the data during a lockdown resulted in variability in interview styles and potential loss of information due to connectivity issues during phone interviews. Additionally, only the Tanzanian researchers delivered full verbatim transcripts, while those from Sierra Leone and Nigeria provided answers in direct context of the structured interview guide without including the superfluous conversation or confusion during connectivity issues. However, all interviews were audio recorded, so the file could be reviewed at any time for clarification. Language barriers were also a factor, with local members of the community and native language speakers conducting the interviews and then translating the findings into English for researchers to analyze. Since the researchers reviewing the data did not conduct the interviews, some points may have been lost in the translation process. However, regular communication between the interviewers and the researchers helped ensure that data was adequately captured and analyzed. A final limitation was that some children were less talkative and offered short responses. This was prevalent among younger and female participants, and this may have slightly skewed the findings.

## 6. Conclusions

Regardless of these limitations, this study offers exciting and rich information about children from low-income countries during a global pandemic. This work provides important lessons to communicate better during this and future crises. Natural and manmade disasters during early childhood development can lead to long-term changes, both positive and negative, in the child’s future. It is important to understand experiences as they are occurring in children’s lives to ameliorate harm.

Future research should continue to examine the lasting ramifications of the pandemic in sub-Saharan countries, taking special care to listen to the voices of children for added insight. While this study only explored three small regions, future work should delve wider and deeper to better understand the generalizability of these results. More research is needed to identify the specific policy implications and recommendations that can help influence child health in sub-Saharan Africa, with this study providing much-needed insight.


*“This event has taught us to take care of our health, and we children are the leaders of tomorrow.”*
(Boy, Age 9, Nigeria)

## Figures and Tables

**Figure 1 ijerph-18-05618-f001:**
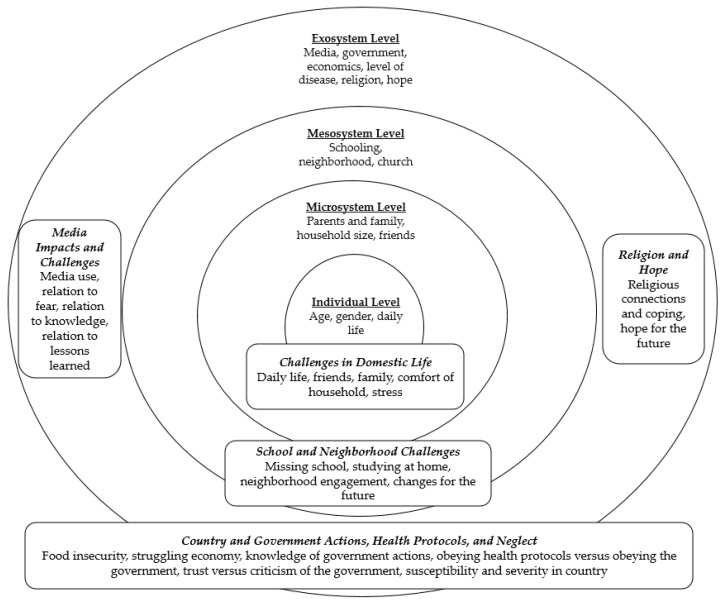
Coding diagram, including major themes and subthemes by ecological systems theory level.

**Table 1 ijerph-18-05618-t001:** Main demographic variables used to stratify results.

Demographic Variables		Nigeria (N = 20)	Sierra Leone (N = 15)	Tanzania (N = 16)	Total (N = 51)
Gender	Boy	11	7	8	26
	Girl	9	8	8	25
Age	9	5	2	4	11
	10	4	3	3	10
	11	5	4	3	12
	12	3	3	3	9
	13	3	3	3	9
	Mean	10.8	11.1	10.9	10.9
Household size	Small (≤6)	8	5	11	24
	Large (7+)	6	10	3	23
	Missing	2	0	2	4
	Range	4–35	4–15	4–10	4–35
	Mean	12.5	8.8	6	9.5
Religion	Muslim	16	7	2	25
	Christian	4	8	14	26

**Table 2 ijerph-18-05618-t002:** Barriers and facilitators by level of Bronfenbrenner’s ecological systems theory, with comparisons by region and demographics.

Factor		Barrier	Facilitator
**Individual Characteristics**			
Gender	Boys	N/A	Same experiences as girls. More talkative. More discussion of the economy. Greater knowledge of symptoms. More hopeful messages for the future.
	Girls	More in tune to worry in the home. Less talkative. Less concerned about economic issues. Less hopeful for the future.	Same experiences as boys. Greater national collectivism.
Age	Older	Most stressed.	Most knowledgeable. Most frequently use media for learning.
	Younger	Least overall issue knowledge. Less praying to manage stress.	Least afraid.Less daily chores to add to burden.
Daily life	Nigeria	High boredom. More misinformation from media.	Chores perceived as fun.
	Sierra Leone	Highest stress levels. High boredom. Chores a burden when interfering with studying. Highest fear spread by media use.	Most praying for comfort and support.
	Tanzania	More misinformation from media.	Most playing. Chores perceived as fun.
	All regions	Media use can spread misinformation.	Bathing common. Media use can provide useful information and lessons learned.
**Microsystem**—**Interpersonal Connections and the Home**			
Parents and family	Nigeria	N/A	Most commonly help at home. Most positive daily family interaction.
	Sierra Leone	Most worried parents. Hiding from family to get peace and quiet to study. Most complaints about not having other children to play with in the household. Highest discomfort in household.	N/A
	Tanzania	Little discussion of family.	N/A
Household size	Large	Most commonly hiding from family. More stress. Greater food insecurity.	More advice to obey health protocols. Greater knowledge of government actions. More praying to cope and messages of religious hope.
	Small	More chores.	More advice to obey the government.
Friends	Sierra Leone	Most children upset that they cannot see friends. A few children seeing their friends despite restrictions.	N/A
**Mesosystem**—**The Community**			
Schooling	Nigeria	Least studying at home.	N/A
	Sierra Leone	N/A	Commonly study at home.
	Tanzania	Most skepticism for the future of school.	Commonly study at home.
	All regions	High worry about missing school.	Use media to study, whether online as in Sierra Leone or through popular media in Tanzania and Nigeria.
Neighborhood	Nigeria	Most frustrated over lack of personal work and access to local markets.	N/A
	Tanzania	Most worry about interactions in the neighborhood.	N/A
Church	Nigeria	N/A	Use religious programs via media.
	Sierra Leone	Most discussion of churches closed.	N/A
	Tanzania	N/A	Use religious programs via media.
**Exosystem**—**The Media, Government, Economy, and Social/Cultural Context**			
Media	Nigeria	Inconsistent knowledge and messaging.	N/A
	Sierra Leone	N/A	High knowledge. Highest media use and access.
	Tanzania	Inconsistent knowledge and messaging.	N/A
	All regions	High connection between high media use and fear.	Teach positive lessons learned for health.
Government	Nigeria	Lack of knowledge of actions. Lowest knowledge, government trust, and discussion of obeying the government.	N/A
	Sierra Leone	N/A	High knowledge of actions.
	Tanzania	Lack of knowledge of actions. Government secrecy.	High national collectivism. High trust in government.
	All regions	Relationship between knowledge/communication and trust/obeying the government. Lack of provisions, leading to criticism.	Obey health protocols.
Economics	Nigeria	Food insecurity.	N/A
	Sierra Leone	Food insecurity.	N/A
	Tanzania	N/A	Government reopening the economy.
	All regions	Struggling economy.	N/A
Level of disease	All regions	Inconsistent messaging and communication.	Take disease seriously with higher communication. Perceived to be less severe at home than abroad.
Religion	Muslim	Food insecurity. Economic struggles. Larger households, leading to too many chores and no quiet time to study. Critical of government. More time seeing friends when in violation of health orders.	Adherence to health protocols. Larger households, leading to more supportive time with family. More time seeing friends for social support.
	Christian	Less time enjoying family.	Trust in government. Obey the government. Daily studying.
Hope	Nigeria	More skepticism and uncertainty about the future.	Express positive messages for the future.
	Tanzania	More skepticism and uncertainty about the future.	Express positive messages for the future.
	All regions	N/A	Value the importance of prayer.

## Data Availability

Information about the interviews and transcripts are available upon request from the corresponding author.

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
