# Peer review of "“People Will Continue to Suffer If the Virus Is Around”: A Qualitative Analysis of Sub-Saharan African Children’s Experiences during the COVID-19 Pandemic"

_ijerph, 2021, doi:10.3390/ijerph18115618_

Round 1
Reviewer 1 Report
In lines 174-180, the authors provide a list of interview questions that were asked of the children. None of the questions listed include a question related to different experiences for boys and girls during the Pandemic. However, on line 244, the authors state “when the interview directly asked participants if they felt boys and girls had difference experiences…”. How structured were the interviews? It seems that there were more questions asked of the children than are listed in lines 174-180. It would be helpful to know exactly what was asked of the children and if the interviews were structured or not.
Author Response
Comment: In lines 174-180, the authors provide a list of interview questions that were asked of the children. None of the questions listed include a question related to different experiences for boys and girls during the Pandemic. However, on line 244, the authors state “when the interview directly asked participants if they felt boys and girls had difference experiences…”. How structured were the interviews? It seems that there were more questions asked of the children than are listed in lines 174-180. It would be helpful to know exactly what was asked of the children and if the interviews were structured or not.
Response: Thank you for pointing this out. The interviews were indeed structured, and interviewers adhered to the interview guide with some flexibility to probe further into a subject when necessary. The methods now have added detail about “adhering to the structured interview guide” and include some additional questions that were posed. This, however, does not include every question asked on the interview guide, and that is reflected in the revisions in lines 172-180.
Reviewer 2 Report
This is the article review for the paper entitled, “People will continue to suffer if the virus is around”: A qualitative analysis of sub-Saharan African children’s experiences during the COVID-19 pandemic” (ijerph-1198721). This paper examines the experience of 9 to 12-year-old individuals from sub Saharan Africa and their reactions to living through the COVID-19 pandemic. Examining the pandemic through the eyes of youth provides a unique perspective to understand how their lives have changed. This novel paper provides important information and could be strengthened if the following issues could be addressed:
Introduction
Page 3, lines 123-24: Is this paper also addressing coping? or just thinking and behaviors? When one is speaking of facilitators and barriers, as in Table 2, it begs the question, Barriers to what? Facilitators of ...coping? Adaptive thinking and behaviors? Please clarify.
Page 3, lines 133-135 “How did children’s experiences and concerns vary by gender, age, household size, religion, and country?” This last question may be more amenable to a mixed methods approach utilizing some form of a quantitative comparison, which would help the reader make sense of differences between genders, regions, religions, etc. As it stands now (see comments in the results section) there are multiple quantitative comparisons being made, but not a good way of making judgments about which ones are truly significant.
Page 5, line 22-212: What was the nature of this training? were there manipulation checks to ensure interviewer stayed true to the manual?
Page 5, lines 218-220: Are these themes part of the analysis? If there were preconceptions on part of the interviewer, how was this information handled once discovered? Is there a plan to publish this information?
Results
In general, well the results are interesting, the manuscript would benefit from a careful review to include results in which participants shared their perceptions of how things have changed during times of COVID pandemic. Decreasing the overall amount and frequency of numerical comparisons within the results section would help decrease reader confusion as to the nature of this study, i.e., whether this is a qualitative versus quantitative study. Flint making more summary statements without including numbers for comparison could also sharpen the manuscript and help it be more readable. Perhaps there are higher order themes that the reader could take away, because as it is written now, there are so many details in the paper which detract from the clarity of the results.
Perhaps a couple of tables comparing some key differences to make a larger point could be helpful. Additionally, doing a mixed methods design would be a powerful method for examining the differences between countries or a couple of variables of interest that were raised through the qualitative analysis. Providing so many details without any method for determining the significance of those differences is just confusing for the reader.
Page 7, Table 2: These descriptors are somewhat confusing. Does "most complaints" really refer to "majority of kids" in the interview from Sierra Leone complain about not having peer playmates? or is this meant to be comparative across regions, i.e. children and families in Sierra Leone had the most complaints of lack of peers compared to families from Tanzania and Nigeria? please clarify
Page 9, line 264-265: This statement is but one example, but the reader is left wondering how boredom is different between pre-COVID and COVID times? In order to make sense of how things are different under COVID conditions, some sort of comparison group or control condition would be necessary, .e.g., how much boredom was experienced pre-covid? If no comparison is available, then simply stating the comments that participants made as to their perceptions of life is different would make the case powerfully. It may be less confusing to leave out most of the comparisons between regions and other characteristics as it just becomes very complex, like keeping a three-way interaction effect in one’s mind.
Page 9, line 281-282. This is a useful and helpful perspective because it is the expression of how life has changed for them, which is what the article purports to discuss.
Page 20, lines 792-794: Lack of full transcripts for interviews in 2/3 of the sample is a very significant limitation and should be mentioned in the methodology section!
Overall, this paper highlights a very important issue-namely, the perceptions of how life has changed for youth ages 9 to 13 during a global pandemic. The inclusion of information from this region of the world is also very important to include in the academic literature. Gathering all of the interviews, transcription and coding of this information is highly valuable to the reader and the author should be commended for the amount of work and effort taken to distill these results down. As noted above, the inclusion of even higher order themes and distillation of comments made by the participants, while removing many of the numeric comparisons, could make the paper far more readable. Finally, were their recommendations for future studies suggested by the interpreters and the teams on the ground in sub Saharan Africa? It strikes me as a reader, that they may have very specific ideas and suggestions for how to improve the lives of individuals living there in comparison to those in Western and developed Nations that are at great distance from the day-to-day lives of individuals in sub Saharan Africa.
Author Response
Comment: This is the article review for the paper entitled, “People will continue to suffer if the virus is around”: A qualitative analysis of sub-Saharan African children’s experiences during the COVID-19 pandemic” (ijerph-1198721). This paper examines the experience of 9 to 12-year-old individuals from sub Saharan Africa and their reactions to living through the COVID-19 pandemic. Examining the pandemic through the eyes of youth provides a unique perspective to understand how their lives have changed. This novel paper provides important information and could be strengthened if the following issues could be addressed:
Introduction: Page 3, lines 123-24: Is this paper also addressing coping? or just thinking and behaviors? When one is speaking of facilitators and barriers, as in Table 2, it begs the question, Barriers to what? Facilitators of ...coping? Adaptive thinking and behaviors? Please clarify.
Response: Thank you for pointing this out. While this paper discusses coping strategies used during the pandemic, the main focus is perceptions, thinking, and behaviors. This clarification was added to lines 123-124, and again to lines 230-231. Barriers and facilitators refer to those identified by children in relation to public health crises like the pandemic that interfere with their lives and health, and are a reflection of their perceptions and thinking.
Comment: Page 3, lines 133-135 “How did children’s experiences and concerns vary by gender, age, household size, religion, and country?” This last question may be more amenable to a mixed methods approach utilizing some form of a quantitative comparison, which would help the reader make sense of differences between genders, regions, religions, etc. As it stands now (see comments in the results section) there are multiple quantitative comparisons being made, but not a good way of making judgments about which ones are truly significant.
Response: Thank you for this thought on the methodology of the paper. This study was conducted as a qualitative inquiry, with the idea of making comparisons across gender, age, household size, religion, and country in an exploratory review of comments and themes from the interviews. This is commonly done in qualitative work to give larger context to the findings and allow the reader to make comparisons more easily. See Reference 27: Creswell JW, Poth CN. Qualitative Inquiry & Research Design: Choosing among Five Approaches; 2018. However, for added clarity, more complex numerical comparisons and references have been removed throughout the results (particularly those directly comparing one group to the other, i.e. X of X versus X of X).
Comment: Page 5, line 22-212: What was the nature of this training? were there manipulation checks to ensure interviewer stayed true to the manual?
Response: We’ve added some additional clarification in lines 215-219 to capture this process. Interviewers and researchers walked through the interview guide together, and interviewers practiced and pretested the guide as well. Additionally, all sessions were audio-recorded to ensure fidelity throughout the data collection process.
Comment: Page 5, lines 218-220: Are these themes part of the analysis? If there were preconceptions on part of the interviewer, how was this information handled once discovered? Is there a plan to publish this information?
Response: Thank you for your question. Yes, the themes presented in the results and Figure 1 are those that emerged throughout the data analysis process. Peer debriefing and reflexivity were used throughout the process to control bias. However, this process did not reveal any concerns or issues worth publishing or noting. We have no plans to publish this information, as it was an iterative process between coders, and there is nothing of consequence to the work to report.
Comment: Results: In general, well the results are interesting, the manuscript would benefit from a careful review to include results in which participants shared their perceptions of how things have changed during times of COVID pandemic. Decreasing the overall amount and frequency of numerical comparisons within the results section would help decrease reader confusion as to the nature of this study, i.e., whether this is a qualitative versus quantitative study. Flint making more summary statements without including numbers for comparison could also sharpen the manuscript and help it be more readable. Perhaps there are higher order themes that the reader could take away, because as it is written now, there are so many details in the paper which detract from the clarity of the results.
Response: Thank you again for your thoughts on the methodology and presentation of this work. Your points are well taken. That said, we find this to be a matter of style in qualitative work, as it is common for tabulations and information on the frequency of comments to be directly mentioned to provide context for comparisons being made. We feel this gives the reader an ability to make comparisons more easily, and adds richness to the manuscript.
See the following:
Creswell JW, Poth CN. Qualitative Inquiry & Research Design: Choosing among Five Approaches; 2018.
Dittmar, H., & Drury, J. (2000). Self-image - is it in the bag? A qualitative comparison between ``ordinary'' and ``excessive'' consumers. Journal of Economic Psychology, 21, 109-142.
Chen, J. C., et al. (2019). Perceptions about e-cigarette flavors: A qualitative investigation of young adult cigarette smokers who use e-cigarettes. Addiction Research & Theory, 27(5), 420-428. doi:10.1080/16066359.2018.1540693
Henry, H. K., & Borzekowski, D. L. (2015). Well, that’s what came with it. A qualitative study of U.S. mothers’ perceptions of healthier default options for children’s meals at fast-food restaurants. Appetite, 87, 108-115. doi:10.1016/j.appet.2014.12.201
We felt it was important to be consistent throughout the manuscript in how this data is presented. However, we have removed some of the numerical references throughout to help further focus the results and address your comment, particularly those that directly compare numbers across groups (X of X versus X of X).
Comment: Perhaps a couple of tables comparing some key differences to make a larger point could be helpful. Additionally, doing a mixed methods design would be a powerful method for examining the differences between countries or a couple of variables of interest that were raised through the qualitative analysis. Providing so many details without any method for determining the significance of those differences is just confusing for the reader.
Response: We hesitate to add more tables into this paper, and our study was not designed with a mixed methods approach in mind. Tabulations provided are given to allow the reader to make comparisons more easily and put the results into context, and this is presented based on the five emerging themes across the four levels of the Ecological Model as presented in Figure 1. As mentioned above, this is something we find to be a stylistic choice that varies across qualitative work. However, we have removed some of the numerical comparisons to help ensure clarity of the results and improve readability, as mentioned above.
Comment: Page 7, Table 2: These descriptors are somewhat confusing. Does "most complaints" really refer to "majority of kids" in the interview from Sierra Leone complain about not having peer playmates? or is this meant to be comparative across regions, i.e. children and families in Sierra Leone had the most complaints of lack of peers compared to families from Tanzania and Nigeria? please clarify
Response: Thank you for pointing this out. The descriptors in the table compare to the other regions or levels of a demographic variable. For example, when the factor is gender, “more discussion of the economy” in boys means that boys more frequently discussed this than girls. In the example you give, “most complaints” from Sierra Leone means more in Sierra Leone than in Tanzania or Nigeria. This has been clarified in the title of the table.
Comment: Page 9, line 264-265: This statement is but one example, but the reader is left wondering how boredom is different between pre-COVID and COVID times? In order to make sense of how things are different under COVID conditions, some sort of comparison group or control condition would be necessary, .e.g., how much boredom was experienced pre-covid? If no comparison is available, then simply stating the comments that participants made as to their perceptions of life is different would make the case powerfully. It may be less confusing to leave out most of the comparisons between regions and other characteristics as it just becomes very complex, like keeping a three-way interaction effect in one’s mind.
Thank you for this thought process. All questions were asked in the context of how things were different during the pandemic, and additional clarification on this has been added to the methods (lines 174-182) as well as to the results (line 230, 275, etc.). All statements are clearly distinguished as the child’s perceptions of life, and we feel the comparisons between regions allow the reader to make preliminary assessments as to how things differed for children in different areas.
Comment: Page 9, line 281-282. This is a useful and helpful perspective because it is the expression of how life has changed for them, which is what the article purports to discuss.
Response: Thank you. All quotes and comments throughout are framed as an expression of life during the pandemic and how it has changed, as referenced above.
Comment: Page 20, lines 792-794: Lack of full transcripts for interviews in 2/3 of the sample is a very significant limitation and should be mentioned in the methodology section!
Response: Thank you for this comment. We have added some clarification to the methods in lines 214-224, as well as to the limitations in lines 802-808. Transcripts for 2/3 of the interviews were not significantly incomplete; instead, they were put directly in the context of the interview guide to delete extra language, largely the consent process and connectivity issues. Additionally, we had audio-recordings of each interview, and therefore there was no data loss. We referred to these recordings as needed for clarification and to ensure data fidelity.
Comment: Overall, this paper highlights a very important issue-namely, the perceptions of how life has changed for youth ages 9 to 13 during a global pandemic. The inclusion of information from this region of the world is also very important to include in the academic literature. Gathering all of the interviews, transcription and coding of this information is highly valuable to the reader and the author should be commended for the amount of work and effort taken to distill these results down. As noted above, the inclusion of even higher order themes and distillation of comments made by the participants, while removing many of the numeric comparisons, could make the paper far more readable. Finally, were their recommendations for future studies suggested by the interpreters and the teams on the ground in sub Saharan Africa? It strikes me as a reader, that they may have very specific ideas and suggestions for how to improve the lives of individuals living there in comparison to those in Western and developed Nations that are at great distance from the day-to-day lives of individuals in sub Saharan Africa.
Response: Thank you for the kind words and constructive feedback. We agree that this work presents an important contribution to the literature. As discussed above, we feel that the numerical comparisons add value to the comparisons being made and give the reader additional context. This is commonly done in qualitative analyses without discussion of statistical significance or quantitative interactions. The final recommendations, results, and conclusions (as well as the entire manuscript) was reviewed and approved by in-country teams to ensure the recommendations were reasonable and true to the lives of those living in sub Saharan Africa. We look forward to sharing these results and recommendations more broadly to advance future research in the field and inform policy and health officials on the ground.
Reviewer 3 Report
It is a rigorous, interesting and necessary investigation, which focuses on a topical issue, such as COVID-19, and on a vulnerable population such as children in 3 countries in sub-Saharan Africa (Sierra Leone, Nigeria and Tanzania). Understanding how these children perceive the health crisis caused by COVID-19 (what do they know about the disease, what do they know about protection measures, etc.) is a topic from which many lessons can be learned. That said, I have some concerns about Bronfenbrenner's theory.
INTRODUCTION
My only concern is related to the different levels considered in your research on Bronfenbrenner's Ecological Systems Theory. I would like the authors to explain why they did not analyze the four layers proposed by Bronfenbrenner and traditionally used in research. For example:
Bronfenbrenner proposed that the developing child is surrounded by layers of relationships like a set of nested Russian dolls (1979, p. 3). The inner circle, which he calls the microsystem, describes each setting in which the child has direct, face-to-face relationships with significant people such as parents, friends, and teachers. This is where students live their daily lives and this is where they develop. Ordinarily, there are crossrelationships between these small settings – parents talk to teachers, for example – and these lateral connections are called the mesosystem (1979, p. 25). Beyond this is an outer circle of people who are indirectly involved in the child‟s development, such as the parents‟ employers, family health care workers, or central school administrators; this is called the exosystem (1979, p. 25). Bronfenbrenner also described a macrosystem (the prevailing cultural and economic conditions of the society) (Leonard, 2011,p.6)
Leonard, J. (2011). Using Bronfenbrenner’s ecological theory to understand community partnerships: A historical case study of one urban high school. Urban Education, 46(5), 987-1010.
METHODOLOGY AND ANALYSIS OF RESULTS
The qualitative methodology has been applied rigorously and clearly. The analysis is very well structured and responds to the objectives of the research. It is based on The Grounded Theory, which is one of the most applied theories for qualitative discourse analysis. Nothing to say about it.
DISCUSSION, RECOMMENDATIONS AND LIMITATIONS
The discussion is consistent with the results and I agree with the limitations identified by the authors.
The only problem that I identify is that some of the recommendations have to do with structural and / or cultural problems that are very difficult to modify.
Author Response
Comment: It is a rigorous, interesting and necessary investigation, which focuses on a topical issue, such as COVID-19, and on a vulnerable population such as children in 3 countries in sub-Saharan Africa (Sierra Leone, Nigeria and Tanzania). Understanding how these children perceive the health crisis caused by COVID-19 (what do they know about the disease, what do they know about protection measures, etc.) is a topic from which many lessons can be learned. That said, I have some concerns about Bronfenbrenner's theory.
Introduction: My only concern is related to the different levels considered in your research on Bronfenbrenner's Ecological Systems Theory. I would like the authors to explain why they did not analyze the four layers proposed by Bronfenbrenner and traditionally used in research. For example:
Bronfenbrenner proposed that the developing child is surrounded by layers of relationships like a set of nested Russian dolls (1979, p. 3). The inner circle, which he calls the microsystem, describes each setting in which the child has direct, face-to-face relationships with significant people such as parents, friends, and teachers. This is where students live their daily lives and this is where they develop. Ordinarily, there are cross relationships between these small settings – parents talk to teachers, for example – and these lateral connections are called the mesosystem (1979, p. 25). Beyond this is an outer circle of people who are indirectly involved in the child’s development, such as the parents‟ employers, family health care workers, or central school administrators; this is called the exosystem (1979, p. 25). Bronfenbrenner also described a macrosystem (the prevailing cultural and economic conditions of the society) (Leonard, 2011,p.6)
Leonard, J. (2011). Using Bronfenbrenner’s ecological theory to understand community partnerships: A historical case study of one urban high school. Urban Education, 46(5), 987-1010.
Response: Thank you for your kind words about our work, and for bringing the Leonard reference to our attention. It seems that researchers apply Bronfenbrenner’s Ecological Systems Theory in different ways across the literature, and we recognize that. In this study, we cite Bronfenbrenner directly for his original definitions of the individual, microsystem, mesosystem, and exosystem levels, and we follow the application in a recent IJERPH paper:
Reference 25. Martínez-Andrés M, Bartolomé-Gutiérrez R, Rodríguez-Martín B, Pardo-Guijarro MJ, Garrido-Miguel M, Mar-tínez-Vizcaíno V. Barriers and Facilitators to Leisure Physical Activity in Children: A Qualitative Approach Using the Socio-Ecological Model. Int J Environ Res Public Health. 2020;17(9):3033. doi:10.3390/ijerph17093033.
We also use the four levels as used explicitly in public and behavioral health, which typically starts with the individual, then moves to local, community, and social/cultural levels:
Reference 26. Hovell MF, Wahlgren DR, Gehrman CA. The Behavioral Ecological Model: Integrating Public Health and Behavioral Science. In: Emerging Theories in Health Promotion Practice and Research: Strategies for Improving Public Health. 2002.
We feel that based on the way this theory is applied in public health, or categorization of the four levels is appropriate.
Comment: Methodology and Analysis of Results: The qualitative methodology has been applied rigorously and clearly. The analysis is very well structured and responds to the objectives of the research. It is based on The Grounded Theory, which is one of the most applied theories for qualitative discourse analysis. Nothing to say about it.
Response: Thank you for this comment. We greatly appreciate it and were rigorous in our methodology and approach.
Comment: Discussion, Recommendations, and Limitations: The discussion is consistent with the results and I agree with the limitations identified by the authors. The only problem that I identify is that some of the recommendations have to do with structural and / or cultural problems that are very difficult to modify.
Response: Thank you. We completely agree that many of these recommendations are difficult to accomplish. That said, the recommendations truly came from the perceptions and experiences of the children during the pandemic, and we hope that this work can shed more light on some of these larger issues. We’ve added a comment at the start of the recommendations in lines 707-708 to reflect that.
Reviewer 4 Report
Thanks for this fascinating paper. The research findings should be communicated with the related governments, health authorities and the local media in a way that’s culturally and language appropriate, once it gets published. This will benefit the children, their families, the communities and the sub-Saharan countries.
This exploratory research used an appropriate Ecological Systems Theory, which helped to analyse the experiences of the children interviewed and organise the themes of the collective experiences. The sample size was roughly 15 for each countries involved, which was not large as the research aimed to not only find out the common themes but also the variations at the 4 levels, i.e. individual, micro, meso and exo. I wonder if the authors could clarify if the children interviewed came from different families (note, some families are very large and there might be more than one child qualified for the research).
Minor typos/things to be clarified:
Lines 27-28, it should be noted that some countries have taken an ‘elimination’ strategy, such as New Zealand and China.
Line 139, brackets [ ] around ref 27 are missing.
Line 261, brackets [ ] around ref 26 are missing.
Line 207, was there a reverse translation of the guide completed to make sure the translated guide was accurate (though it needs to be culturally sensitive as well)?
Table 2, I wonder if it helps to have the dotted lines as separators for the rows as it is not easy to read between the three columns: Factor, Barrier and Facilitator.
Line 399, how was ‘high media use’ defined?
Line 431, it would be more appropriate to say ‘the first reported case’.
Lines 567-569, “Participants from larger households (9 of 23, 12 mentions) seemingly more frequently talked about religion than those from smaller households (7 of 24, 10 568 mentions).” Whether it is ‘more’ is not very much evident here, as it is based on small difference and small sample size (no statistical test was performed here anyway).
Line 805, I’d suggest to say ‘during this and future crises’.
Author Response
Comment: Thanks for this fascinating paper. The research findings should be communicated with the related governments, health authorities and the local media in a way that’s culturally and language appropriate, once it gets published. This will benefit the children, their families, the communities and the sub-Saharan countries.
Response: Thank you for this comment and your thorough read of the manuscript. We completely agree, and we plan to disseminate a press release and coordinate the sharing of results with local media and health authorities through our in-country teams as much as possible.
Comment: This exploratory research used an appropriate Ecological Systems Theory, which helped to analyse the experiences of the children interviewed and organise the themes of the collective experiences. The sample size was roughly 15 for each countries involved, which was not large as the research aimed to not only find out the common themes but also the variations at the 4 levels, i.e. individual, micro, meso and exo. I wonder if the authors could clarify if the children interviewed came from different families (note, some families are very large and there might be more than one child qualified for the research).
Response: As you mentioned, the sample size was very much an exploratory sample and is presented as such in the manuscript. Thank you for noting this. No children were sampled from the same household in this study, and we’ve added a clarification in the methods in lines 154-155. The four levels of the Ecological Systems Theory help to contextualize and make sense of these exploratory findings, but the study was never powered with the intention of making broader comparisons across the levels. This structure just adds richness to the analysis and interpretation of the exploratory results, with the hope that the work can draw attention for future research and larger-scale work.
Comment: Minor typos/things to be clarified: Lines 27-28, it should be noted that some countries have taken an ‘elimination’ strategy, such as New Zealand and China.
Response: This comment has been added to the introduction in lines 27-28.
Comment: Line 139, brackets [ ] around ref 27 are missing.
Response: Brackets have been added.
Comment: Line 261, brackets [ ] around ref 26 are missing.
Response: Brackets have been added.
Comment: Line 207, was there a reverse translation of the guide completed to make sure the translated guide was accurate (though it needs to be culturally sensitive as well)?
Response: Thank you for pointing this out. Yes, part of the translation process involved reverse translation for accuracy. The guide was also pre-tested on children by in-country teams as specified and reviewed by local teams to ensure cultural sensitivity. This detail is now all reflected in the minor revisions in lines 208-212.
Comment: Table 2, I wonder if it helps to have the dotted lines as separators for the rows as it is not easy to read between the three columns: Factor, Barrier and Facilitator.
Response: Thank you for this comment. The tables were formatted based on the journal standards, and they will be put into the appropriate form for easy reading by the journal. We agree that either more white space or a dotted line between columns could be helpful.
Comment: Line 399, how was ‘high media use’ defined?
Response: High media use was defined by daily usage, while moderate media use was defined as near daily or multiple times per week. A slight modification to lines 399-400 now clarifies this.
Comment: Line 431, it would be more appropriate to say ‘the first reported case’.
Response: Agreed. This change has been made.
Comment: Lines 567-569, “Participants from larger households (9 of 23, 12 mentions) seemingly more frequently talked about religion than those from smaller households (7 of 24, 10 mentions).” Whether it is ‘more’ is not very much evident here, as it is based on small difference and small sample size (no statistical test was performed here anyway).
Response: Thank you, this adjustment has been made to the language. As is common with qualitative work, tabulations are provided to help give a sense of the frequency of comments and make general comparisons, but no statistical testing was conducted.
Comment: Line 805, I’d suggest to say ‘during this and future crises’.
Response: Agreed. This change has been made.
Round 2
Reviewer 2 Report
This is the article re-review for the paper entitled, “People will continue to suffer if the virus is around”: A qualitative analysis of sub-Saharan African children’s experiences during the COVID-19 pandemic” (ijerph-1198721-revised). This paper examines the experience of 9 to 12-year-old individuals from sub Saharan Africa and their reactions to living through the COVID-19 pandemic. Examining the pandemic through the eyes of youth provides a unique perspective to understand how their lives have changed. This novel paper provides important information and could be strengthened if the following issues could be addressed:
Introduction
Page 3, lines 123-24: Is this paper also addressing coping? or just thinking and behaviors? When one is speaking of facilitators and barriers, as in Table 2, it begs the question, Barriers to what? Facilitators of ...coping? Adaptive thinking and behaviors? Please clarify. This has been addressed.
Page 3, lines 133-135 “How did children’s experiences and concerns vary by gender, age, household size, religion, and country?” This last question may be more amenable to a mixed methods approach utilizing some form of a quantitative comparison, which would help the reader make sense of differences between genders, regions, religions, etc. As it stands now (see comments in the results section) there are multiple quantitative comparisons being made, but not a good way of making judgments about which ones are truly significant. This has been appropriately addressed.
Page 5, line 22-212: What was the nature of this training? were there manipulation checks to ensure interviewer stayed true to the manual? Good clarification.
Page 5, lines 218-220: Are these themes part of the analysis? If there were preconceptions on part of the interviewer, how was this information handled once discovered? Is there a plan to publish this information? Thank you for the clarifications on this point.
Results
In general, well the results are interesting, the manuscript would benefit from a careful review to include results in which participants shared their perceptions of how things have changed during times of COVID pandemic. Decreasing the overall amount and frequency of numerical comparisons within the results section would help decrease reader confusion as to the nature of this study, i.e., whether this is a qualitative versus quantitative study. Finally making more summary statements without including numbers for comparison could also sharpen the manuscript and help it be more readable. Perhaps there are higher order themes that the reader could take away, because as it is written now, there are so many details in the paper which detract from the clarity of the results.
Perhaps a couple of tables comparing some key differences to make a larger point could be helpful. Additionally, doing a mixed methods design would be a powerful method for examining the differences between countries or a couple of variables of interest that were raised through the qualitative analysis. Providing so many details without any method for determining the significance of those differences is just confusing for the reader. Thank you for the references to Creswell and other qualitative research. After consideration, these are fair points and make sense. It also makes the paper more readable by removing many of the numerical comparisons.
Page 7, Table 2: These descriptors are somewhat confusing. Does "most complaints" really refer to "majority of kids" in the interview from Sierra Leone complain about not having peer playmates? or is this meant to be comparative across regions, i.e. children and families in Sierra Leone had the most complaints of lack of peers compared to families from Tanzania and Nigeria? please clarify. Thank you for the clarification.
Page 9, line 264-265: This statement is but one example, but the reader is left wondering how boredom is different between pre-COVID and COVID times? In order to make sense of how things are different under COVID conditions, some sort of comparison group or control condition would be necessary, .e.g., how much boredom was experienced pre-covid? If no comparison is available, then simply stating the comments that participants made as to their perceptions of life is different would make the case powerfully. It may be less confusing to leave out most of the comparisons between regions and other characteristics as it just becomes very complex, like keeping a three-way interaction effect in one’s mind. Thank you for the helpful clarifications.
Page 9, line 281-282. This is a useful and helpful perspective because it is the expression of how life has changed for them, which is what the article purports to discuss.
Page 20, lines 792-794: Lack of full transcripts for interviews in 2/3 of the sample is a very significant limitation and should be mentioned in the methodology section! Thank you for clarifying just what information was missing and highlighting this in the manuscript.
Overall, this paper highlights a very important issue-namely, the perceptions of how life has changed for youth ages 9 to 13 during a global pandemic. The inclusion of information from this region of the world is also very important to include in the academic literature. The work that was done on the original manuscript and now these changes make the paper far more readable. After supporting literature as it relates to qualitative analysis of data, the way the paper is laid out makes more sense and is adequately described. Finally, the removal of all of the quantitative information from the results section makes it more readable as well. Overall, the authors should be commended for this rigorous and well-done study. They have adequately and appropriately addressed each of my concerns as noted above.
Reviewer 3 Report
No more suggestions